# Workplace Assessment Scale: Pilot Validation Study

**DOI:** 10.3390/ijerph191912408

**Published:** 2022-09-29

**Authors:** Eileen Huang, Nicole E. Edgar, Sarah E. MacLean, Simon Hatcher

**Affiliations:** 1Faculty of Medicine, University of Ottawa, 451 Smyth Road, Ottawa, ON K1H 8M5, Canada; 2Clinical Epidemiology Program, Ottawa Hospital Research Institute, 501 Smyth Road, Ottawa, ON K1H 8L6, Canada; 3School of Journalism and Communication, Carleton University, 1125 Colonel By Drive, Ottawa, ON K1S 5B6, Canada; 4Department of Psychiatry, University of Ottawa, 1145 Carling Avenue, Ottawa, ON K1Z 7K4, Canada

**Keywords:** scale design, reliability, validity, first responders, mental health, workplace, occupational stress, humiliation, betrayal

## Abstract

First responders, such as police officers, paramedics, and firefighters are at an increased risk of experiencing negative mental health outcomes compared to the general population. This predisposition can partially be attributed to common occupational stressors, which may provoke strong feelings of betrayal and humiliation. The Workplace Assessment Scale (WAS) was developed as there is currently no appropriate measure to assess such feelings in the first responder population. Initial development of the WAS included a Betrayal Subscale and the Humiliation Subscale, each comprised of 5 Likert scale questions which ask participants to report the frequency at which they experience specific feelings associated with their workplace. This pilot validation study was conducted to determine if there is preliminary evidence to support a large-scale validation study. To determine this, we assessed the internal structure and the convergent, concurrent, and predictive validity of the WAS. Based on 21/22 (95%) participant responses, a factor analysis did not support the two-factor model we anticipated, with only one factor and seven items retained from the original version of the scale. However, the internal consistency of the remaining items was strong. The validity analysis found moderate convergent validity and weak predictive validity based on correlations between the WAS and other psychometric scales. Minimal concurrent validity was noted. Additional research is needed for further analysis and validation of the WAS.

## 1. Introduction

First responders, such as firefighters, paramedics, and police officers, protect the safety and security of the public by being among the first to arrive and provide assistance at a scene of an emergency [1]. Due to the nature of their role, first responders are at increased risk of exposure to potentially psychologically traumatizing events (PPTE), such as sudden or accidental death, serious on-road accidents, and physical assault, where they may be both a witness and a victim [2]. As a result, first responders are at increased risk of negative mental health outcomes compared to the general population. A survey of Canadian public safety personnel, which includes first responders, by Carleton et al. found a positive correlation between high or repeated incidence of exposure to PPTE and the odds of screening positive for post-traumatic stress disorder (PTSD), major depressive disorder (MDD), panic disorder (PD), and generalized anxiety disorder (GAD), establishing a dose–response relationship [2].

While the mental health burden on individuals of workplace exposure to PPTE is an active area of research, there has been little attention paid to how public safety occupational factors contribute to mental disorders. Occupational stressors can be categorized into firstly operational stressors, which stem directly from work-related duties, such as shift work, over-time, work-related injury, and strain of job on work–life balance. Secondly, there are organizational stressors, which relate to the job environment and culture, such as co-worker dynamic, treatment from leadership, staff shortages, barriers to accessing help or insurance, and lack of training [2]. Recent literature suggests that these occupational factors are more often the root cause of psychological injuries among first responders than exposure to PPTE [2,3]. However, there is a substantial gap in knowledge regarding how the culture, policies, and operations of first responder organizations contribute to such stressors. Of specific interest is how mental disorders may be precipitated or worsened by feelings of betrayal and humiliation which develop after first responders feel poorly supported by their organizations, especially following exposure to PPTEs. This has been observed clinically among first responders who have sought care at an Occupation Stress Injury clinic in Ottawa, Canada (see Appendix A for complete vignettes).

In this first vignette, Jane, a first responder who has been with the service for 21 years describes feeling abandoned by her organization after going on sick leave for a post-traumatic stress injury:


*“I have been a first responder passionately serving several communities for over 21 years. For me the ‘iceberg’ moment was one year ago when I took up a new position. I had safety concerns about what I was being asked to do but management did not support my concerns. My supervisor turned up unannounced at work and was relentlessly antagonizing, yelling at me, and berating me in front of my team and the public. I was condemned for my ‘unprofessional behaviour’ and told that I would have to attend a mandatory disciplinary meeting. I was emotionally devastated so I went off on sick leave.*



*The return-to-work process was hostile and difficult. My employer refused to give me any modified duties. I was told I had to return to full duties or not return at all. On my first shift back to work, my employer sent me an email with an “Exit Package” asking me to consider leaving the organization. This sent me into a deep depression, and I lost myself completely. I struggled to even get out of bed. So, once again I was forced to go off on sick leave. Upon my return from this second leave, I was sent an email stating I had to attend a performance review for disciplinary action where I had my integrity and professionalism questioned. I felt immediately vulnerable and felt attacked.*



*Now, here I am off work again feeling completely defeated, worthless, and abandoned. I have lost all the confidence I once had. I feel completely unsafe in my workplace and worry constantly that they will find a way to fire me. I feel like a fool for being so dedicated thinking it was valued and appreciated. So stupid for thinking that my employer would be there to support me. How can first responders possibly continue to fight the daily battles out on the frontlines when there is this lethal battle going on behind the scenes?”*


Similarly, Mark speaks to the mental health impacts of a 20-year career as a first responder. He describes being targeted for dismissal by his employer while he was on sick leave. The resultant feelings of betrayal had a significant impact on his recovery:


*“My story started in April 2001 which was when I started my career as a first responder. I was excited to be part of something that I was missing in my life, a family. The service became my family, and it would consume my life. The brotherhood and sisterhood that I was part of became the most important part of my life.*



*Over the course of 20 years, I worked alongside my colleagues witnessing some of the worst calls I could’ve imagined. I didn’t know it at the time, but the accumulation of these calls would later lead to my Post Traumatic Stress Injury. I became very depressed after 20 years of being on the job, and I was ashamed to admit that I had a problem. PTSD impacted my sleep, I became very angry around people, I started drinking alcohol heavily and I started having a fascination with committing suicide.*



*After years of feeling this way, I went off of work. My employer started coming after me with policy violations. They accused me invalid notebook entries, using a force vehicle for personal use, obtaining unsanctioned car washes, and improper time scheduling. As a result of these accusations, the employer wants to dismiss me from work. I felt targeted, betrayed and hated. My colleagues turned their backs on me. People I was close to wouldn’t even acknowledge me in public. I’ve been in therapy dealing with my PTSI for over a year now. I have accepted the fact that I can’t be a first responder anymore, and I’m good with this decision, but the actions of my organization have had a tremendous negative impact on my rehab.”*


Despite the impact that occupational stressors may have on first responder mental health and well-being, few psychometric instruments have been developed to evaluate organizational betrayal or humiliation. While measures targeting first responder occupational stressors have been developed, such as the Police Stress Survey (1989), the Police Daily Hassles (1993), the Organizational Police Stress Questionnaire (PSQ-Org) (2006), and the Operational Police Stress Questionnaire (PSQ-Op) (2006), these scales were not designed to capture feelings of betrayal and humiliation as they primarily ask participants to identify which stressors have impacted them most [4,5,6]. Additional limitations to these scales include survey burden due to their length as well as the failure to consider the experiences of first responder professions other than police officers [6].

Scales assessing institutional betrayal have also been developed. However, the scales are context specific and inappropriate for use in the first responder population. The Institutional Betrayal Questionnaire (IBQ) was first developed to measure institutional betrayal in individuals who have experienced sexual assault [7]. Since then, the scale has been adapted to assess institutional betrayal in additional settings: IBQ-Health measures institutional betrayal in healthcare systems, IBQ-Climate measures institutional betrayal in school settings, and IBQ-COVID measures experiences regarding institutional protocols and policies regarding COVID-19 [8,9,10]. There is currently no appropriate scale to assess feelings of betrayal, humiliation, or embitterment in the first responder population.

The burden of occupational stressors and the significance of betrayal and embitterment have been highlighted in the literature and by first responders themselves. However, the lack of an appropriate measure to assess the impact of such stressors reveals a clear gap in evidence for the treatment and prevention of mental health disorders in first responders. This study, thus, describes the development and pilot validation of a novel self-report scale to measure workplace satisfaction, focusing specifically on experiences of betrayal and humiliation among the first responder population. The overarching goal of this study was to determine whether preliminary evidence supported the conduct of a large-scale validation study in this population.

## 2. Materials and Methods

### 2.1. Objectives

The current study is a pilot validation study which assessed the content validity, internal structure, convergent validity, concurrent validity, and predictive validity of the novel Workplace Assessment Scale (WAS). The WAS is an instrument we developed to measure workplace satisfaction in the first responder population, motivated by the need expressed by first responders and a gap in the literature as highlighted above. The key objective of this pilot study was to determine whether there was preliminary evidence to support a large-scale validation study in this population.

### 2.2. Phase 1: Instrument Development

#### Content Validity: Theoretical Framework and Item Generation

The development of the WAS was guided by the biopsychosocial formulation model which offers a framework for understanding the influences involved in the development and maintenance of a multitude of psychiatric illnesses [11,12,13,14]. This model considers the four Ps (predisposing, precipitating, perpetuating, and protective factors) in the context of biological, psychological, and social domains. Firstly, predisposing factors are those that make an individual vulnerable to a psychiatric diagnosis. Second, precipitating factors refer to events which trigger the onset of a mental disorder. Third, perpetuating factors are conditions that exacerbate a mental disorder. Lastly, protective factors are influences which counteract the predisposing, precipitating, and perpetuating factors.

The WAS addresses occupational stressors which encompass all four Ps within the social domain of the biopsychosocial formulation model. Examples of how occupational stressors fit within the biopsychosocial formulation model, along with examples of biological, psychological, and social factors, have been outlined in Table 1.

The accumulation of, and repetitive exposure to, organizational stressors may provoke strong feelings of betrayal and humiliation amongst the first responder population, resulting in experiences of institutional betrayal and embitterment. Institutional betrayal refers to the deliberate negligence or failure of a trusted institution to respond appropriately to negative experiences [7,15]. Consequently, the members who rely on such institutions for support, protection, and resources often experience a violation of trust [15]. Embitterment may also result as a consequence of institutional betrayal. It refers to the inclination to undo an experience, reinstate justice, or take revenge after an experience of humiliation, a breach of trust, or injustice [6]. Experiences of institutional betrayal and embitterment have been studied in systems including university campuses where a sexual assault has occurred, in the Canadian medical system, and in the military but never before in first responder organizations [7,16,17,18].

Based on this, we operationalized embitterment as including both feelings of betrayal and humiliation, with a subset of items dedicated to each concept. The final version of the WAS included a total of ten items. For each item, participants are asked to indicate the frequency at which they experienced specific feelings on a Likert scale from 0 to 3 (not at all, occasionally, a lot of the time, or most of the time). Item generation was guided by the Oxford English Dictionary definitions for betrayal and humiliation. Betrayal is defined as a “violation of trust or confidence, an abandonment of something committed to one’s charge” [19]. The betrayal subscale measures feelings of abandonment (Q1, Q5), confidence (Q3, Q9), and trust (Q7). To humiliate is defined as an act “to lower or depress the dignity or self-respect of; to subject to humiliation; to mortify” [20]. The humiliation subscale assesses feelings of anger (Q2), a sense of value (Q4), embarrassment (Q6), humiliation (Q8), and potential mistreatment (Q10). Items included in the final version of the scale are described in Table 2 below.

### 2.3. Phase 2: Scale Validation

To assess the reliability and validity of the WAS, we used standard indicators of scale validity [21,22].

#### 2.3.1. Internal Structure

Based on the theoretical framework described above, we hypothesized that the internal structure of the WAS would be made up of two distinct factors. First, we expected that items related to the disloyalty or deception of the organization to load positively onto a betrayal factor. Second, we expected items related to individual feelings of shame, disgrace, or dishonour to load positively onto a humiliation factor (Hypothesis 1).

#### 2.3.2. Convergent Validity: Relationship with a Standard Measure for Burnout

Given that the WAS was designed to assess workplace (dis)satisfaction, we anticipated a strong correlation with a standard measure of organizational burnout (Hypothesis 2). To explore this hypothesis, the Maslach Burnout Inventory-Human Services Survey (MBI-HSS) was administered [23]. This 22-item questionnaire asks participants to rate the frequency at which they experience feelings of burnout on an 8-point Likert scale ranging from 0 (never) to 7 (every day). The MBI-HSS assesses feelings of emotional exhaustion (e.g., “I feel emotionally drained from my work”), depersonalization (e.g., “I feel I treat some recipients as if they were impersonal objects”), and personal accomplishment (e.g., “I deal very effectively with the problems of my recipients”). Initial validation of the Maslach Burnout Inventory (MBI) showed good internal consistency with a Cronbach’s alpha of 0.83 (frequency) and 0.84 (intensity). Coefficients for test–retest reliability for all subscales surpassed significance (*p* < 0.001) [23]. The internal consistency of the MBI-HSS scales ranged from 0.62 to 0.93 in the current study (Table 3).

#### 2.3.3. Concurrent Validity: Relationship with Work-Related Measures

The academic literature has established a connection between occupational stress and other work-related measures, such as productivity and absenteeism. For example, in a sample of 770 registered nurses, Labrague et al. found that those working in a toxic environment reported lower job contentment, higher stress levels, frequent absenteeism, and higher intent to leave the nursing profession [24]. Similarly, in a sample of police officers, Nisar and Rasheed found that occupational stress was negatively associated with career satisfaction, performance of one’s duties, and the extent to which participants went “above and beyond” in their role [25]. If the WAS is measuring workplace (dis)satisfaction as intended, it should logically be correlated with other work-related measures (Hypothesis 3). To assess this, the Workplace Productivity and Impairment Questionnaire: General Health (WPAI:GH) was administered [26]. This is a 6-item self-report that measures the impact of an individual’s health concerns, including mental health, over the previous seven days. The first portion of the scale assesses absenteeism, the extent of work time missed, while the latter portion assesses productivity, the impairment of professional and non-work-related activities. The WPAI:GH shows good test-rest reliability (less than 5% difference over 12 months) and good convergent validity across multiple chronic health conditions [26,27]. As can be seen in Table 3, the study examined the relationship between overall scores on the WPAI:GH as well as responses to the absenteeism (Question 2) and impact on productivity (Question 5) items.

#### 2.3.4. Predictive Validity: Relationship to Other Mental Health Outcomes

While understandings of the link between exposure to PPTE and the development of subsequent mental health disorders is an evolving area of research, there is preliminary evidence that suggests that this link can be explained by occupational stressors, as described above. The WAS taps into many organizational stressors, including leadership, treatment by management, and culture. Given this, there may reasonably be a link between organizational dissatisfaction and the presence of mental health disorders (Hypothesis 4). To explore this further, a battery of mental health outcomes were assessed (see Table 3 for overview), including depression, anxiety, post-traumatic stress disorder (PTSD), and alcohol and drug misuse.

### 2.4. Sample and Participant Recruitment

This secondary analysis was conducted using data which was originally collected as part of an observational study that assessed the acceptability of a first responder mental health clinic and associated mental health team in Ottawa, Canada. From September 2020 to June 2021, participants for the observational study were recruited from the First Responder Clinic at The Ottawa Hospital and from the Ottawa Paramedic Service. All study participants were at least 18 years old, fluent in English and/or French, and were employees of either Ottawa Fire Service, Ottawa Paramedic Service, or Ottawa Police Service. All participants provided verbal informed consent and have been included in the current secondary analysis.

### 2.5. Statistical Analysis

IBM Statistical Package for the Social Sciences (SPSS) Version 28.0 was used for all statistical analyses conducted in this study. To characterize the participant sample, descriptive statistics are presented as mean (SD) for continuous variables and absolute values with percentages for categorical variables.

To assess the internal structure of the WAS, exploratory principal components factor analysis with varimax rotation was used. As discussed above, the total number of participants in the original observational study (*n* = 22) determined the sample size for this secondary analysis. However, following the work of MacCallum et al. (1999), factor analysis in small sample sizes is justified when communalities for each item are above 0.6 [28]. In determining which items to retain for each factor identified, we used Comrey and Lee’s cut-off of 0.63 [29]. A factor was considered stable if it had at least 4 items with a factor loading of at least 0.6 [30]. Index sub-scale variables were created for each factor identified. Internal consistency of the WAS questionnaire and its sub-scales were determined using Cronbach’s alpha. A threshold of 0.70 was used to indicate strong reliability [31]. Item means, corrected item-to-total correlations (i.e., the correlation between an individual item and the total score without that item), Cronbach’s alpha if item deleted, and inter-item correlations were also obtained from the reliability analysis. A corrected item-total correlation of <0.30 was the threshold value used to indicate an item as a candidate for deletion [32]. List-wise deletion was used in both the principal components factor analysis and the reliability analysis.

Convergent, concurrent, and predictive validity of the WAS were assessed using correlational analysis. Pearson’s correlation (parametric variables) and Spearman’s rank correlation coefficient (non-parametric variables) were used to assess the relationships between baseline WAS scores and scores on the burnout, work-related, and mental health outcomes identified in Table 3. Weak, moderate, and strong correlations were indicated as ≤0.39, 0.40–0.69, and ≥0.70, respectively, [33]. Pair-wise deletion was used to calculate all correlation statistics. An overall 5% type I error level was used to infer statistical significance.

### 2.6. Bias

Numerous precautions were taken to reduce common method bias throughout the course of the study. The WAS scale does not feature any questions regarding the severity of organizational stressors that the respondents may have experienced. Additionally, workplace stressors were not a part of any other scale which was administered during the baseline visit. This eliminates sources of item priming effects and measurement context effects which may contribute to common method bias [34]. Respondents were also reminded that they were able to skip questions when answering the scales to limit effects of item ambiguity and social desirability bias.

Nonetheless, there were also some sources of common method bias throughout the study. For instance, approximately half of the sample was recruited from the first responder mental health clinic described above. Social desirability bias may have influenced the manner in which these participants responded to the survey if they recognized that the study doctor was also their treating psychiatrist. However, the team took precautions to remedy this source of bias by scheduling study visits separately from visits with the doctor and were conducted by research members only. Beyond this, all self-report measures were completed independently by participants using an Electronic Data Capture System (EDCS).

## 3. Results

### 3.1. Participant Characteristics

A total of 22 participants were enrolled in the study. The demographic data of the sample are outlined in Table 4. The gender distribution of study participants was nearly balanced, with 54.5% of participants identifying as male and 45.5% of participants as female. No participants identified as transgender or gender non-conforming. The age range of participants was 23–63 years, with a mean age of 40.5 years (SD 9.6 years). It is important to note that all participants self-identified as White. Participants primarily belonged to Ottawa Paramedic Services (19/22, 86.4%), followed by Ottawa Fire Services (2/22, 9.1%), and Ottawa Police Services (1/22, 4.5%). The majority of study participants (15/22, 68.2%) were working full-time at entry into the study, while 31.8% (7/22) were on leave. No participants had less than a college diploma. The WAS was completed by 21/22 (95%) participants. One participant, who was on leave during the relevant timeframe, did not complete the WAS as they felt that the questions were not applicable to them at this time. Of the 21 participants who did complete the WAS, questions 1–7 and 10 were answered 100% of the time, while question 8 was only responded to 86% of the time (18/21) and question 9 90% of the time (19/21).

### 3.2. Internal Structure

While the WAS was completed by 21/22 (95%) participants, only 18 of the 21 participants (86%) provided responses to all items of the questionnaire. The exploratory factor analysis identified two components, cumulatively accounting for 80.5% of the variance in the model, with Component 1 and Component 2 accounting for 48.45% and 32.05% of the variance, respectively. Initial eigenvalues, extraction sums of squared loadings, and rotation sums of squared loadings are described in Table 5.

Item communalities and factor loadings are described in Table 6. Component 1 contained both betrayal (*n* = 4) and humiliation items (*n* = 3). Factor loadings for this component ranged from 0.786 to 0.909 (Table 6). All items were retained as they were above the cut-off score of 0.63. Component 2 also contained both betrayal (*n* = 1) and humiliation (*n* = 2) items. Factor loadings for this component ranged from 0.781 to 0.970. However, this factor was not retained as it did not have the required four items with factor loadings above 0.6 necessary to be considered stable. Thus, items 8–10 were not included in subsequent analyses.

The seven items that were retained demonstrated strong internal consistency, with a Cronbach’s alpha of 0.952. The reliability statistics are outlined in Table 7, including item means, corrected item-total correlations, and Cronbach’s alpha if an item is deleted. Inter-item correlations are also described in Table 8. All of the retained items displayed high corrected-item total scores, ranging from 0.805–0.895, and high inter-item correlations, ranging from 0.605 to 0.880. Thus, none of the items were considered for deletion.

### 3.3. Convergent Validity

Correlations between the WAS and other outcomes are described in Table 9. A moderate positive relationship between the WAS and the emotional exhaustion sub-scale of the MBI-HSS was noted (ρ = 0.608, 95% CI = 0.212–0.832, *p* = 0.041). However, there was no statistically significant relationship between scores on the WAS and scores on the depersonalization (ρ = 0.241, *p* = 0.320) or personal accomplishment (ρ = −0.306, *p* = 0.249) sub-scales of the MBI-HSS.

### 3.4. Concurrent Validity

There no statistically significant relationships noted between the WAS and other work-related measures, including absenteeism and productivity (Table 9).

### 3.5. Predictive Validity

A moderate positive relationship between the WAS and baseline depressive symptoms, as measured by the PHQ-9, was noted (ρ = 0.479, 95% CI = 0.047–0.760, *p* = 0.033). However, this was not maintained at the 12-week time point. There was no statistically significant relationship between WAS scores and the other mental health outcomes included in the study (Table 9).

## 4. Discussion

The objective of this pilot validation study was to evaluate the internal structure and preliminary validity of the WAS. In terms of the internal structure of the scale, we hypothesized that there would be two clear components identified during the factor analysis: a betrayal and a humiliation sub-scale (Hypothesis 1). Instead, the first seven items of the WAS loaded highly onto one component and the remaining three items on a second component. However, the second component did not meet the threshold of a stable factor (a minimum of four items with loadings of at least 0.6) [30]. This suggests that items 8–10 will need to be expanded in future iterations of the scale to include additional related items to form the basis of a stable sub-scale. Internal consistency of the remaining seven items (Q1–Q7) appeared to be strong, with strong corrected item-total and moderate to strong inter-item correlations. No items were identified for deletion.

The WAS demonstrated moderate convergent validity with a statistically significant relationship to a standard indicator of burnout (MBI-HSS, Emotional Exhaustion Sub-Scale) (Hypothesis 2). Given this, our preliminary results support that the WAS is a valid measure to assess feelings of humiliation and betrayal related to organizational stressors.

However, the concurrent validity of the WAS was minimal (Hypothesis 3). No significant relationships were noted between the WAS and absenteeism or productivity measures despite their described relevance in the academic literature [24,25]. The lack of correlation between absenteeism, productivity, and total WAS scores may potentially be attributed to a combination of factors. The use of a single question on the WPAI:GH to measure absenteeism and productivity, respectively, may lack sensitivity. Additionally, first responders are often subjected to strict work leave policies which do not permit unverified absences of more than 1–2 days in the timeframe outlined in question 2 of the WPAI:GH. As such, the measurement of absenteeism as measured by one question of a general questionnaire may not be sufficient in this setting. A multi-dimensional assessment of absenteeism may be necessary for further analysis. Since this study was carried out during the COVID-19 pandemic, patterns of absence may have also been altered due to additional factors such as high-risk contact, infection with COVID-19, or coverage for colleagues off with COVID-19. These confounding factors may also be contributing to the lack of correlation between absenteeism and WAS scores.

Similarly, the predictive validity of the WAS was weak (Hypothesis 4). Despite previous work by Carleton et al. describing strong associations between occupational stressors and negative mental health outcomes [2], our study only found a moderate relationship to depressive symptoms at the baseline time point and this was not maintained at the 12-week time point and no relationships with any other measures were noted. One potential explanation for this is that the survey completed by Carleton et al. used self-report measures, which likely overestimate the prevalence of common mental disorders [35,36]. Another potential explanation for this discrepancy is the recruitment of a sample which may not be actively experiencing mental health disorders. While our study used both self-report and observer-rated assessments, over half of our sample was experiencing mild or no symptoms of depression, anxiety, or PTSD. Only three participants had potentially hazardous drinking behaviour and no participants reported substance misuse. Repeating the correlational analysis with a larger sample of participants who are experiencing at least moderate levels of severity, as measured by an observer-rated assessment, for each disorder may be more likely to replicate the finding by Carleton et al. and to identify any relationship of the WAS with mental disorders.

The WAS was developed as there is currently no validated scale to characterize the impact of occupational stressors on the mental health of the first responders. Our results suggest that the WAS may be capturing complex feelings of betrayal and humiliation related to their organization and occupation which have not yet been described in the current literature. Additional research is needed to further understand these constructs and to ensure that they are accurately captured in the WAS.

### 4.1. Strengths

There are multiple strengths of the WAS and of this study. The WAS is the first questionnaire to quantify how occupational stressors collectively translate into more complex feelings of betrayal and humiliation as previous scales primarily focused on quantifying the negative impact of various stressors. While prior studies have only focused on the police, we were able to recruit from paramedic and fire services in addition to police, increasing the generalizability of our findings [4,5,6]. To date, the WAS is the shortest occupational stressors-related questionnaire with a total of 7 items. This minimizes survey burden for respondents, which has previously been associated with low response rates in self-reports and specifically identified as a limitation of occupational questionnaires [6,37]. However, caution should be used to ensure that the quality of the content is not sacrificed solely for the purpose of having a brief measure [37]. Additionally, our analysis of concurrent and predictive validity investigated potential correlations between total WAS scores and other psychiatric outcomes, which was a missing component in the validation of other occupational stress scales for the first responder population [6].

### 4.2. Limitations

This study also has several limitations. First, is the small sample size. Recruitment for this study took place during the COVID-19 pandemic when first responders were facing high workload, unpredictable duties and procedures, and low morale. These factors certainly contributed to the challenges of recruiting a larger, and possibly more diverse, sample. Second, calculations were made with the assumption that data is missing at random, but a small sample size makes it difficult to confirm that this is the case. A small sample size also limited the generalizability of our results. This said, the sample size of the factor analysis was justified by extremely high communalities between the items of the WAS [28]. The external validity of our study was further limited as all study participants are Caucasian, which is inconsistent with the current diversity of Ottawa’s first responder services. There is also an over-representation of paramedic staff and an under-representation of staff from the Ottawa fire and police services. Lastly, face validity was not measured in this pilot study.

### 4.3. Future Directions

The results of this pilot study will be used to inform the design of a full validation study of the WAS. Future studies will aim to recruit a larger sample size with greater ethnic diversity and a more equal distribution of first responder occupations. It may also be useful to conduct the full validation with a national population of first responders to ensure that the measure is generalizable across the broader population. With a more representative sample, reliability and validity analyses of the WAS will be repeated with the inclusion of additional items related to Q8–Q10 to create a more stable second factor. Additional analyses may include a more thorough assessment of content validity, involving experts in the area of first responder mental health, the inclusion of an assessment of face validity by obtaining participant feedback about the WAS, test–retest reliability, and more extensive correlational analysis by collecting WAS scores at multiple follow-up time points.

## 5. Conclusions

Our aim for this pilot validation study was to determine whether there was sufficient preliminary evidence to conduct a large-scale validation study of the WAS, as a tool to measure how occupational stressors may provoke feelings of betrayal and humiliation in first responders. Findings from our analysis suggest that the WAS demonstrates strong reliability and moderate convergent validity. Weak predictive validity was also noted, with minimal correlations between WAS and other psychiatric scales. Based on this, we will be moving forward with a larger study to assess the validity of this instrument more fulsomely. Ultimately, we hope that the WAS could be incorporated into routine clinical screening and management for the mental healthcare of first responder patients. We also hope that the use of this scale will encourage first responder staff and leadership to recognize the importance of organizational wellbeing as well as the need and opportunity for impactful change in their organizations.

## Figures and Tables

**Table 1 ijerph-19-12408-t001:** Overview of the 4 Ps of the biopsychosocial formulation.

	Biological	Psychological	Social (Focusing on Occupational Stressors)
Predisposing	Genetic vulnerability Age Gender Race	Socioeconomic status Adverse childhood events (e.g., Childhood neglect)	Staffing shortages Overtime/shift work Poor workplace culture
Precipitating	Substance misuse Onset of medical condition	Negative/maladaptive thoughts	Exposure to PPTEs Lack of organizational support
Perpetuating	Underlying chronic illness	Self-destructive coping mechanisms	Ongoing workplace conflict
Protective	Above average intelligence	Adaptive coping mechanisms	Strong social supports Healthy work environment

**Table 2 ijerph-19-12408-t002:** WAS items and scoring.

Item	Text	Sub-Scale	Response Categories			
			Most of the Time	A Lot of the Time	Occasionally	Not at All
1	I feel that my organization will always support me	Betrayal	0	1	2	3
2	I feel angry at my organization	Humiliation	3	2	1	0
3	I feel that what my organization says publicly about values and how it treats its members are different	Betrayal	3	2	1	0
4	I feel that my organization values me	Humiliation	0	1	2	3
5	I feel let down by the organization I work for	Betrayal	3	2	1	0
6	I feel embarrassed by how my organization has treated me	Humiliation	3	2	1	0
7	I trust the senior management of my organization to do the right thing	Betrayal	0	1	2	3
8	I feel my organization is making an example of me to deter others	Humiliation	3	2	1	0
9	I feel I have been blamed unfairly by my organization for problems at work	Betrayal	3	2	1	0
10	I find it difficult to think about anything else apart from how my organization has treated me	Humiliation	3	2	1	0

**Table 3 ijerph-19-12408-t003:** Outcome measures overview.

Measure	Scale	Items	Current Study α	Item Example
Burnout				
Emotional Exhaustion	MBI-HSS	9	0.93	I feel emotionally drained from work.
Depersonalization	MBI-HSS	5	0.62	I feel I treat some recipients as if they were impersonal objects.
Personal Accomplishment	MBI-HSS	8	0.77	I deal very effectively with the problems of my recipients.
Work-Related Outcomes				
Workplace Productivity and Impairment	WPAI:GH	6	-	During the past seven days, how many hours did you miss from work because of any other reason, such as vacation, holidays, time off to participate in this study?
Absenteeism	WPAI:GH, Q2	1	-	During the past seven days, how many hours did you miss from work because of your health problems?
Impact on Productivity	WPAI:GH, Q5	1	-	During the past seven days, how much did your health problems affect your productivity while you were working?
Mental-Health Outcomes				
Depression	PHQ-9	9	0.78	Little interest or pleasure in doing things.
Anxiety	GAD-7	7	0.90	Feeling nervous, anxious, or on edge.
PTSD	PCL-5	20	0.93	Repeated, disturbing, and unwanted memories of the stressful experience?
Alcohol Misuse	AUDIT	10	0.76	How often do you have a drink containing alcohol?
Drug Misuse	DAST-10	10	*	Have you used drugs other than those required for medical reasons or have you misused prescription drugs?
Mental Well-Being	WEMWBS	14	0.89	I’ve been feeling optimistic about the future.

* Cronbach’s alpha could not be calculated for the current study as very few participants endorsed any of the scale items.

**Table 4 ijerph-19-12408-t004:** Participant characteristics (*n* = 22).

Demographics	Frequency *n* (%)
Gender	
Male	12 (54.5)
Female	10 (45.5)
Age	
Range	23–63
Mean, SD	40.50 (9.6)
Marital Status	
Common Law	7 (31.8)
Married	8 (36.4)
Single	3 (13.6)
Divorced	2 (9.1)
Widowed	1 (4.5)
Prefer not to answer	1 (4.5)
Ethnicity	
Caucasian	22 (100)
Education	
College diploma	16 (72.7)
University/Bachelor’s degree	6 (27.3)
Service	
Ottawa Paramedic Service	19 (86.4)
Ottawa Fire Service	2 (9.1)
Ottawa Police Service	1 (4.5)
Employment Status	
Full-time	15 (68.2)
Currently on leave	7 (31.8)

**Table 5 ijerph-19-12408-t005:** Variance explained by WAS components.

Component	Initial Eigenvalues			Extraction Sums of Squared Loadings			Rotation Sums of Squared Loadings		
	Total	% of Variance	Cumulative %	Total	% of Variance	Cumulative %	Total	% of Variance	Cumulative %
1	6.358	63.58	63.58	6.358	63.58	63.58	4.85	48.45	48.45
2	1.692	16.92	80.50	1.692	16.92	80.50	3.205	32.05	80.50

**Table 6 ijerph-19-12408-t006:** WAS item communalities and factor loadings.

Item	Communalities		Component Matrix		Rotated Component Matrix	
	Initial	Extraction	Component 1	Component 2	Component 1	Component 2
1. I feel that my organization will always support me	1.000	0.771	0.779	−0.405	0.871	0.111
2. I feel angry at my organization	1.000	0.843	0.917	−0.057	0.786	0.475
3. I feel that what my organization says publicly about values and how it treats its members are different	1.000	0.827	0.760	−0.500	0.909	0.022
4. I feel that my organization values me	1.000	0.700	0.796	−0.256	0.800	0.243
5. I feel let down by the organization I work for	1.000	0.844	0.887	−0.238	0.865	0.310
6. I feel embarrassed by how my organization has treated me	1.000	0.764	0.872	0.056	0.685	0.543
7. I trust the senior management of my organization to do the right thing	1.000	0.768	0.869	−0.115	0.779	0.400
8. I feel my organization is making an example of me to deter others	1.000	0.869	0.716	0.597	0.249	0.898
9. I feel I have been blamed unfairly by my organization for problems at work	1.000	0.944	0.606	0.760	0.066	0.970
10. I find it difficult to think about anything else apart from how my organization has treated me	1.000	0.720	0.719	0.452	0.333	0.781

**Table 7 ijerph-19-12408-t007:** WAS item descriptives and item-total statistics.

Item	Item Mean (SD)	Corrected Item-Total Correlation	Cronbach’s Alpha If Item Deleted
1. I feel that my organization will always support me	2.14 (0.854)	0.806	0.940
2. I feel angry at my organization	1.57 (0.926)	0.887	0.933
3. I feel that what my organization says publicly about values and how it treats its members are different	2.05 (1.024)	0.805	0.940
4. I feel that my organization values me	2.19 (0.750)	0.811	0.941
5. I feel let down by the organization I work for	1.86 (1.014)	0.895	0.932
6. I feel embarrassed by how my organization has treated me	1.38 (1.244)	0.818	0.944
7. I trust the senior management of my organization to do the right thing	2.24 (0.768)	0.834	0.939

**Table 8 ijerph-19-12408-t008:** WAS inter-item correlations.

	Item 1	Item 2	Item 3	Item 4	Item 5	Item 6	Item 7
Item 1							
Item 2	0.714						
Item 3	0.736	0.814					
Item 4	0.659	0.772	0.704				
Item 5	0.833	0.784	0.778	0.695			
Item 6	0.605	0.843	0.653	0.776	0.759		
Item 7	0.784	0.713	0.621	0.699	0.880	0.737	

**Table 9 ijerph-19-12408-t009:** Correlations between WAS and outcome measures.

	Baseline					Week 12				
	N	Mean (SD)	Median (IQR)	r/ρ	*p*	N	Mean (SD)	Median (IQR)	r/ρ	*p*
Burnout (MBI-HSS) *										
Emotional Exhaustion	**19**	**31.68 (13.25)**	**30.00 (20)**	**0.608**	**0.006**	-	-	-	-	-
Depersonalization	19	14.68 (6.83)	16 (13)	0.241	0.320	-	-	-	-	-
Personal Accomplishment	16	4.25 (7.30)	34 (12)	−0.306	0.249	-	-	-	-	-
Work-Related Outcomes										
Work Productivity and Impairment (WPAI:GH)	19	6.79 (4.64)	5 (7)	0.293	0.237	9	3.78 (3.35)	3 (2)	−0.420	0.260
Absenteeism (WPAI:GH, Question 2)	18	10.22 (16.86)	0 (18)	−0.117	0.645	13	14.77 (26.90)	0 (18)	0.008	0.980
Productivity (WPAI:GH, Question 5)	19	3.37 (3.27)	3 (4)	0.179	0.477	9	1.67 (1.94)	1 (1.25)	−0.624	0.072
Mental Health Outcomes										
Depression (PHQ-9)	**21**	**10.62 (4.76)**	**11 (8.5)**	**0.479**	**0.033**	16	9.56 (5.97)	8.5 (7.5)	−0.261	0.347
Anxiety (GAD-7)	22	8.68 (5.68)	7 (11)	0.093	0.689	16	10.69 (5.47)	10 (10.5)	0.103	0.714
PTSD (PCL-5)	20	29.25 (16.47)	28.5 (22)	0.132	0.590	17	29.65 (17.12)	25 (25.5)	−0.030	0.912
Alcohol Misuse (AUDIT)	18	3.67 (3.01)	2.5 (4.25)	−0.047	0.855	15	3.8 (2.96)	3 (4)	−0.115	0.683
Drug Misuse (DAST-10)	18	19.36 (0.63)	19 (1)	−0.384	0.195	9	0.67 (1.12)	0 (1.5)	−0.247	0.521
Mental Wellbeing (WEMWBS)	22	42.95 (8.15)	42.5 (7.75)	−0.279	0.221	17	39.65 (17.11)	41 (14)	0.245	0.360

* The MBI-HSS was only administered at the baseline time point.

## Data Availability

The data presented in this study are available on request from the corresponding author. The data are not publicly available due to ethical restrictions.

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
