# Peer review of "Workplace Assessment Scale: Pilot Validation Study"

_ijerph, 2022, doi:10.3390/ijerph191912408_

Round 1
Reviewer 1 Report
The work is valuable, but I believe it needs to strengthen the following aspects:
1. The construction of the scale should be argued from a theoretical perspective.
2. The convergent validity process needs to be supported to give validity to the results. This implies theoretical support. Although you indicate that you use Pearson's correlation, but my question is, under what parameters?
3. Regarding the Likert scale, it is recommended that it has symmetry and equidistance. “A good Likert scale, as above, will present symmetry of Likert items about a middle category that have clearly defined linguistic qualifiers for each category (Hair et al., 2017:35). If you took a different approach, please mention and argue theoretically.
Reviewer 2 Report
Thank you for the opportunity to read the manuscript “Workplace Assessment Scale: Pilot Validation Study”. The authors present a very interesting study related to a particular context. Although I believe the manuscript has potential, I considered it should be rejected because of the sample size. The authors justify their sample size however, I do not see that the manuscript content with such a small sample justifies its publication. Authors could increase the sample size, improve the validation process to be more robust, and try the publication.
In addition, I would like to highlight other comments regarding the manuscript:
- The name of the proposed scale is not clear enough. It is difficult to understand what the content is, and the nature of the scale.
- Also, some theoretical information is missing in the “introduction” . From my point of view, the introduction should be smaller, and the authors could add a section regarding to the theoretical background. In that section, add what we know about this construct, what it means (some definitions are presented in the materials…) and references to the relationship with some leadership constructs (toxic leadership?) because authors refer that this could be a leadership phenomenon.
- Items of the WAS should be presented in the document and not in the supplementary files.
- Some item examples of the other scales also should be presented
- There is no reference to Exploratory or / and Confirmatory Factor Analysis (probably because of the sample size) but this is a mandatory analysis in a Validation Study (even when it is a pilot one)
- The author did not mention and precautions or remedies regarding common method bias
- In my opinion, the presented vignette of jane testimonial does not add any value to the manuscript
I hope my comments add value to the authors work.
Best Wishes
Round 2
Reviewer 2 Report
Thank you for the opportunity to read, once again, the manuscript. I consider that authors integrate most of my comments and suggestions, improving the overall merit of the manuscript.
However, the sample size is still a major concern. Because the authors responded in a very complete way to all my comments I will leave the decision to the editor.
Thank you